# Individualized Medication Review in Older People with Multimorbidity: A Comparative Analysis between Patients Living at Home and in a Nursing Home

**DOI:** 10.3390/ijerph19063423

**Published:** 2022-03-14

**Authors:** Núria Molist-Brunet, Daniel Sevilla-Sánchez, Emma Puigoriol-Juvanteny, Lorena Bajo-Peñas, Immaculada Cantizano-Baldo, Laia Cabanas-Collell, Joan Espaulella-Panicot

**Affiliations:** 1Geriatric Department, Hospital Universitari de la Santa Creu de Vic, 08500 Vic, Spain; lbajo@chv.cat (L.B.-P.); icantizano@chv.cat (I.C.-B.); jespaulella@chv.cat (J.E.-P.); 2Central Catalonia Chronicity Research Group (C3RG), Centre for Health and Social Care Research (CESS), Universitat de Vic—University of Vic-Central University of Catalonia (UVIC-UCC), 08500 Vic, Spain; danielsevillasanchez@gmail.com; 3Pharmacy Department, Parc Sanitari Pere Virgili, 08023 Barcelona, Spain; 4Epidemiology Department, Hospital Universitari de Vic, 08500 Vic, Spain; epuigoriol@chv.cat; 5Tissue Repair and Regeneration Laboratory (TR2Lab), Research Group, University of Vic, 08500 Vic, Spain; 6Institut Català de la Salut (ICS), 08551 Tona, Spain; lcabanas.cc.ics@gencat.cat; 7Geriatric and Palliative Care Department, Hospital Universitari de Vic, 08500 Vic, Spain

**Keywords:** inappropriate prescription, nursing homes, polypharmacy

## Abstract

(1) Background: aging is associated with complex and dynamic changes leading to multimorbidity and, therefore, polypharmacy. A periodic medication review (MR) in frail older people leads to optimizing medication use. The aims of the study were to perform a comparative analysis of the impact of place of residence (own home versus nursing home) in a cohort of older patients on the characteristics of the baseline therapeutic plan and characteristics of the therapeutic plan after an MR; (2) Methods: Study with paired pre- and post-MR data based on person-centred prescription, with a follow-up assessment at three months. Patients who lived either in their own home or in a nursing home were recruited. We selected patients of 65 years or more with multimorbidity whose General Practitioner identified difficulties with the prescription management and the need for an MR. Each patient’s treatment was analysed by applying the Patient-Centred Prescription (PCP) model; (3) Results: 428 patients. 90% presented at least one inappropriate prescription (IP) in both settings. In nursing homes, a higher number of implemented optimization proposals was detected (81.6% versus 65.7% (*p* < 0.001)). After the MR, nursing-home patients had a greater decrease in their mean number of medications, polypharmacy prevalence, therapeutic complexity, and monthly drug expenditure (*p* < 0.001); (4) Conclusions: PCP model detected a high number of IP in both settings. However, after an individualized MR, nursing-home patients presented a greater decrease in some pharmacological parameters related to adverse events, such as polypharmacy and therapeutic complexity, compared to those living at home. Nursing homes may be regarded as a highly suitable scenario to carry out a periodic MR, due to its high prevalence of frail people and its feasibility to apply the recommendations of an MR. Prospective studies with a robust design should be performed to demonstrate this quasi-experimental study along with a longitudinal follow-up on clinical outcomes.

## 1. Introduction

In recent decades, the increasingly ageing population has brought with it a high prevalence of chronic diseases. As chronic diseases are associated with functional impairment and disability, a large number of older adults require daily assistance and are often placed in nursing homes [1].

Older patients with multimorbidity (the coexistence of two or more chronic conditions) often present frailty criteria, which is a common clinical syndrome that carries an increased risk of poor health outcomes [2]. Consequently, they usually require greater medical care and, as a result, polypharmacy (the concurrent use of five or more medications [3]) is common among them.

Current evidence has associated multimorbidity and frailty with poorer health outcomes, such as impaired physical and cognitive function, falls and hip fractures, usually in the context of greater exposure to polypharmacy and the use of anticholinergic and sedative drugs [3,4,5]. Furthermore, frailty and polypharmacy increase the risk of receiving inappropriate prescriptions [3,6,7,8,9]. This increases the risk of suffering adverse drug events (ADE) related to pharmacokinetic and pharmacodynamic changes by drug-drug interaction (DDI) associated with these patients’ multiple comorbidities [4,5,8]. Several studies show that medications are often inappropriately prescribed to older patients [7,8,10], especially among frail individuals with polypharmacy and comorbidities [7]. According to the evolution of each individual patient, medications that may previously have been considered appropriate can become inappropriate. This depends on the progression of a chronic condition or the appearance of a new diagnosis that implies a change in the patient’s primary care aim [11].

Given the marked vulnerability of frail patients, there is concern and evidence they may not benefit from intensive management of chronic conditions in the same way as study populations do. Therefore, it becomes essential to ensure that the benefit of treatment outweighs the harm it may do to very vulnerable patients, in whom the risk of side effects may be particularly high [3,12].

As a result, according to current evidence, a medication review (MR) is considered essential in frail patients, especially in those with polypharmacy [13,14]. It is recommended to carry out a periodic MR using a specific tool to consider parameters such as quality of life, functionality, main care goal, and life expectancy [13]. According to the Pharmaceutical Care Network Europe (PCNE), an MR is defined as a structured evaluation of a patient’s medicines to optimize their use and improve health outcomes. This fact entails detecting medication-related problems and recommending interventions to ensure individualized prescription [14,15].

In addition, to make proposals to ensure individualized prescription, it is considered mandatory to bear in mind the patient’s values, barriers, and limitations concerning taking medications, strategies to self-manage medications, and health literacy [14]. The Patient-Centered Prescription Model (PCP) is a type of MR that integrates these concepts and combines clinical judgement and scientific evidence in a pragmatic and systematic approach. There are different studies applying the PCP model, carried out in acute care hospitals, intermediate care hospitals, and nursing homes, that show the capacity to identify inappropriate prescriptions (IP), reduce polypharmacy prevalence, and improve medication adherence in older patients [16,17,18,19,20].

This scenario becomes especially relevant in nursing homes due to the high proportion of patients with multimorbidity, functional and cognitive impairment, and consequently, frailty [21]. In this context, establishing a geriatric assessment to improve polypharmacy in nursing homes becomes essential [22].

### Study Objectives

The objectives of the study were to perform a comparative analysis of the impact of place of residence (own home vs. nursing home) in a cohort of older patients (COP-cohort [21]) in (i) baseline situation; (ii) characteristics of the baseline therapeutic plan: prevalence of polypharmacy, number of (IP), medication complexity, anticholinergic and sedative burden and, monthly drug expenditure (MDE); (iii) the same characteristics of the therapeutic plan after a medication review based on the application of a patient-centred prescription model.

## 2. Materials and Methods

### 2.1. Design

This was a quasi-experimental pre-post study of a cohort of patients who lived either in their own home or in a nursing home (Community Older Patients cohort (COP cohort)) [21], in the county of Osona, a mixed urban-rural district in Catalonia (Spain), with a three-month follow-up. Data were collected from June 2019 to October 2020. The study was conducted in three Primary Care Centres and three different nursing homes.

Some of the results of the same study with COP cohort, using different range of research data, have been previously published [21].

Inclusion criteria: patients of 65 years or older who suffered multimorbidity (two or more morbidities) whose General Practitioner had identified difficulties with prescription management and the need for an MR by a consultant team, made up of a geriatrician and a clinical pharmacist.

Exclusion criteria: patients in their probable last hours or days of life [23].

Ethics approval: We obtained verbal informed consent from patients or their main caregivers. Afterwards, we included the patient’s verbal informed consent in their electronic health record. The study was approved by the Scientific Ethics Committee, of each site: (1) FORES (Fundació d’Osona per la Recerca i l’Educació Sanitàries), under reference number 2019-106/PR237; (2) IDIAP Jordi Gol, under reference number 19/206-P; (3) Fundació Catalana d’Hospitals, under reference number CEI 20/23.

### 2.2. Data Collection

Data was collected at the beginning of the study, before the MR, and after three months (post-MR).

Personal data: Age, gender, and place of residence.

Functional data: Dependence/independence for medication management. Dependence/independence for basic activities of daily living (Barthel Index (BI) [24]).

Medical data: (i) Morbidities (from the expanded diagnostic clusters within the Johns Hopkins University ACG system [25]) and age-adjusted Charlson Index [26]; (ii) Dementia diagnosis, as stated in the medical records, and the degree of deterioration was established in accordance with GDS (Global Deterioration Scale) [27]; (iii) Blood pressure levels available in the last year; (iv) a patient was considered to suffer depressive syndrome, anxiety syndrome or other psychiatric disorders when they took a specific medication treatment. (v) Geriatric syndromes falls (2 or more falls during the last six months), dysphagia, pain (2 or more conventional analgesics or major opioids), pressure ulcers, constipation (when the patient took a chronic laxative), insomnia (when the patient took specific sedative medication), malnutrition (weight loss of at least 5% in last six months), incontinence or delirium (when the patient presented with delirium in last six months which required neuroleptic treatment).

Analytical data: (i) calcium; (ii) glycosylated haemoglobin (HbA1c) if available during the last year. Frailty index (FI): It was measured using the Frail-VIG index [28]. FI was categorized as: (i) no frailty if FI < 0.20; (ii) mild frailty if FI 0.20–0.35; (iii) moderate frailty if FI 0.36–0.50; (iv) severe frailty if FI > 0.50.

Pharmacological data: The number of chronic medications was recorded for at least six months before the MR (baseline) and after a three-month follow-up (post-MR). Polypharmacy was analysed at baseline and post-MR, and it was categorized in three different degrees: (i) no polypharmacy (0–4 medications); (ii) moderate polypharmacy (from 5 to 9 medications), and excessive polypharmacy (10 or more medications) [29] at baseline and post-MR.

Medication Regimen Complexity Index (MRCI) [30,31]: This is the most widely-used validated instrument for assessing the complexity of medication regimens. The index consists of 65 items grouped into three sections with different scores assessing the complexity of dosage form, dosing frequency, and additional directions. The result is a continuous scale in which higher scores indicate greater regimen complexity. It was analysed at baseline and post-MR, and it was categorized into three different degrees: (i) low complexity if 0–19.99; (ii) moderate complexity if 20–39.99; (iii) high complexity if ≥40.

Anticholinergic and sedative burden: Anticholinergic and/or sedative risk exposure for regularly-scheduled long-term medications was assessed using the drug burden index (DBI). This index was developed according to pharmacologic principles to examine the association between medication use and physical and cognitive performance [32]. It was analysed at baseline and post-MR, and the DBI was categorized into three different degrees: (i) low if it was 0–0.99; (ii) moderate if it was 1–1.99; (iii) high if it was ≥2.

Identification of end-of-life patients (EOL patients) (using NECPAL CCOMS-ICO© tool criteria [33]): These were patients considered to be in the final months or year of their life. The criteria to identify them as EOL patients were based on: (i) their primary care physician; (ii) advanced illness criteria [33]; (iii) frail-VIG index > 0.50.

Individual main therapeutic goal: According to their baseline situation, a therapeutic goal was established through a consensus with the patient (or caregiver in cases of incapacity), their usual healthcare team, and the consultant team: (i) survival in patients with a fit baseline situation; (ii) functional in patients with an intermediate baseline situation; and (iii) symptomatic control in the most vulnerable patients.

Mortality: At three months follow-up, the comparative study between pre- and post-pharmacological data was carried out with the total number of patients alive.

Monthly Drug Expenditure (MDE): The cost of prescribed medications for each patient was collected (in euros (€)).

Periodic meetings were scheduled between different primary care teams (General Practitioner and nurse) and a consultant team (a geriatrician and a clinical pharmacist). Before the meeting, the primary care team identified the older patients with multimorbidity who may be having difficulties with prescription management. Once the patient was identified, the primary care team informed the patient (or main caregiver in cases of incapacity) and requested their informed consent to carry out an MR along with a consultant team.

If the patient consented, a global assessment of the patient and an MR were carried out by the consultant team as a first step (before the interdisciplinary meeting). Then, during the meeting, the patient’s global baseline assessment and the MR were made (the Barthel index evaluation was calculated, and the global Deterioration Scale and frailty index were evaluated). All prescribed medications were recorded (and the related pharmacological indices were calculated: MRCI, DBI and monthly drug expenditure) and proposals were made for optimizing the pharmacological plan. Finally, the teams reached an agreement on the proposals for optimizing individual medication. Patients’ data were incorporated successively during the different meetings

After the meeting, the agreed medication proposals were shared between the primary care team and the patient (or main caregiver in cases of incapacity) during a conventional visit. Additionally, in accordance with the decisions made between them, the accepted proposals were recorded in the electronic prescription.

At three months of follow-up, the electronic prescription was reviewed by the consultant team, and the proposals recorded in the electronic prescription were considered to be accepted and were transferred to a TeleForm document.

### 2.3. Medication Review

Each patient’s treatment was analysed by applying the Patient-Centred-Prescription (PCP) model [21]. This is a systematic 4-stage process, carried out by an interdisciplinary team formed by the patient’s General Practitioner and the nurse with a consultant team (a geriatrician and a clinical pharmacist). The model centers therapeutic decisions on the patient’s global assessment (comprehensive geriatric assessment (CGA), the calculation of the FI [34]), and the resulting individual main therapeutic goal. These decisions were taken in conjunction with the patient or with the main caregiver in cases of incapacity according to the baseline situation (Figure 1).

#### Criteria Used to Determine IP

The following recommendations were used to identify IP in the most prevalent chronic conditions [21]:

Patients at the end of life (according to NECPAL CCOMS-ICO© [33]): the indication of medications aimed at prolonging survival was reassessed. Medications for primary prevention were evaluated for potential discontinuation and those for secondary prevention were individualized in accordance with patient goals [35].

Type 2 Diabetes Mellitus (T2DM): to optimize hypoglycaemic therapy two important proposals were considered: (i) Therapeutic intensity criteria: taking American Diabetes Association (ADA) guidelines as our basis [36,37], we established a maximum HbA1c target for each patient profile (Table 1). (ii) Qualitative criteria regarding drug prescription to consider inappropriate prescription: The prescription of sulphonylureas (SU) was considered inappropriate due to their high risk of hypoglycaemia [37,38]. Patients with doses of metformin were not adjusted for renal failure [37]. And the use of insulins associated with the highest risk of hypoglycaemic episodes (short-acting insulins, mixtures, and postprandial use) was considered inappropriate, except in justified cases [37].

Hypertension and cardiovascular therapy: it is recommended less intensive control in people with multimorbidity, especially in cases of dementia or limited life expectancy [39]. We proposed measures for pharmacological adjustment in end-of-life patients whose mean systolic blood pressure (SBP) was under 130 mmHg in the last year [35].

Dyslipidaemia: statins are not recommended in end-of-life patients [35], regardless of the indication, especially for primary prevention. In addition, withdrawal of lipid-lowering medication was suggested for people who had total cholesterol (TC) lower than 150 mg/dL, given that it is a malnutrition marker [40,41].

Mental health and Dementia: the recommendations of the European Association of Palliative Care were followed. They define a different therapeutic objective in patients with dementia according to the evolutionary stage of their pathology [42]. Regarding chronic antipsychotic treatment, the progressive decrease in doses was proposed in people who had not had behavioural disorders in the last 3–6 months [35,43].

Pain: in accordance with Beers/STOPP criteria, the following proposals were made [38,44,45]: (i) Tricyclic antidepressants to treat neuropathic pain were avoided, due to their anticholinergic effects. (ii) Non-steroidal anti-inflammatory drugs (NSAID) were recommended at the lowest dose and for the shortest time possible. (iii) Weak opioids such as tramadol and codeine were recommended only at low doses. (iv) Major opiates, such as morphine or oxycodone, should always be combined with a laxative. (v) Meperidine is not recommended because of its anticholinergic potential.

Osteoporosis: it is recommended to withdraw treatment with calcium supplements (except in cases where symptomatic hypocalcaemia is being treated), vitamin D, or antiresorptive drugs [35] in patients identified as at the end-of-life.

Figure 2 shows the number of patients analysed pre- and post-MR.

### 2.4. Sample Size

To calculate sample size, IP in the overall older frail population was estimated at 43% [46]. With a 95% confidence level and 5% accuracy, it was estimated that a minimum of 352 patients should be included.

### 2.5. Data Analysis

Statistical analysis was performed with IBM SPSS Statistics v27.0 statistical software (IBM, Armonk, NY, USA). The results for categorical variables were expressed as absolute and relative frequencies and results for continuous variables were presented as means and standard deviations (SD). The Chi-square test or Fisher’s exact test (in 2 × 2 tables where the expected frequencies were lower than 5) were used to evaluate the relationship between qualitative variables and the patients’ place of residence. Students’ *t*-test was used to analyse the relationship between quantitative variables and the patients’ place of residence. Statistical significance was declared when the value of *p* was less than 0.05.

## 3. Results

Overall, 210 patients living in their own home (3 refusals (1.4%)) and 218 living in a nursing home (2 refusals (0.91%)) were recruited. Table 2 outlines the COP-cohort’s baseline demographic, clinical, functional, and cognitive data according to their place of residence, and Table 3 lists the baseline pharmacological data.

Nursing homes showed a higher proportion of women and an increased prevalence of patients with functional dependence and cognitive impairment. Overall, the degree of frailty was also higher among institutionalized patients, and they presented an increased mean number of geriatric syndromes. EOL patients were specially represented in nursing homes (51.8% versus 20.0% at home) (*p* < 0.05).

Pre- and post-pharmacological data were compared according to the place of residence (Table 4). Although nursing-home patients took fewer medications, there were no significant differences regarding IP according to the place of residence (*p* = 0.209).

In nursing homes, at three months of follow-up, a higher number of accepted optimisation proposals was detected (474 out of 581 initial proposals were implemented (81.6%) versus 393 out of 598 in patients living at home (65.7%) (*p* < 0.001)). Consequently, after the MR, nursing-home patients had a greater decrease in their mean number of medications (*p* < 0.001). MRCI also showed a greater decline among nursing-home patients (*p* < 0.001). However, after the MR, DBI decreased in both settings without significant differences (Table 4). Overall, the median of the prescription cost decreased by up to 20.1% (from 57.61 € to 46.03 €/month after the MR) with the application of the PCP model. However, nursing-home patients showed a larger difference between pre- and post-MR cost (from 51.81 € to 36.46 €/month versus from 71.24 € to 60.59 €/month in patients living at home) (*p* < 0.001).

## 4. Discussion

The two groups are unsurprisingly different, as is described in literature [19,47]. However, it is remarkable that after the MR a greater number of proposals for pharmacological optimization had been implemented to nursing-home patients than to patients living at home, and, consequently, there was a greater decline in data related to polypharmacy, MRCI, and MDE among nursing-home patients. Two main reasons could justify these results: First, the higher prevalence of frailty patients in nursing homes and, also, in an end-of-life situation, which are the situations with the most evidence of medication optimization. Secondly, the close relationship between physicians, nurses, caregivers, and family members that exist in nursing homes, could facilitate the progressive implementation of pharmacotherapeutic proposals and enable a closer clinical follow-up. Thus, the frailer the patients are, the more improvement of pharmacological parameters could be expected after the application of an individualized MR. This is a relevant result to encourage the implementation of a periodic MR in nursing homes because it could lead to considering nursing homes as a suitable setting to apply a periodic MR carried out by an interdisciplinary team.

It should be noted that more than half of the people in nursing homes were identified as end-of-life, a higher proportion than that observed in other national and international studies [48,49]. The finding of a higher number of comorbidities in patients living at home is not shared with the rest of the literature [48]; it can be related to less clinical data collection among nursing-home patients than people living at home. However, this data supports the accepted concept that the biggest difference between patients living in nursing homes and those living at home is not the number of diagnoses but the fact that they all present dependence and frailty, which are the characteristics that most determine a high degree of difficulty to continue living at home [48].

Despite nursing-home residents having a lower average of chronic medications, they presented the same proportion of IP as patients living at home. Other studies conducted in the nursing home setting have also detected a prevalence of around 90% of IP [19]. Regarding DBI, the highest DBI detected in nursing homes does coincide with the rest of the studies [48] and can be explained by the higher prevalence of patients with some degree of cognitive impairment.

Currently, several studies that applied an MR separately in both community-dwelling patients and nursing-home residents reveal results aligned with this project, with a reduction of chronic medications, MRCI, and DBI [20,50,51].

According to the literature, the most notable results of this study may be those related to the involvement of the patient (or main caregiver in cases of incapacity), the interdisciplinary approach, and the presence of the clinical pharmacist in the decision-making process [52,53].

Overall, it is important to highlight that the PCP model leads to an increase in prescription quality, resulting in a lower cost. Thus, the PCP model could be a value-based care tool, especially in nursing homes.

The current study has some limitations. Firstly, the two samples are not strictly comparable, because it was not the primary objective of the study. Thus, the results should be treated with a degree of caution. The COP-cohort was created to analyse the baseline situation of older patients with multimorbidity and the results of applying MR. However, after the whole process, the differences in pharmacological outcomes detected according to the place of residence proved to be significant. Secondly, the results are focused on the process and its pharmacological parameters and not on clinical outcomes, which could be of greater interest. Finally, the expenditure analysis did not consider the cost of the professionals involved in optimizing medication.

As this is a quasi-experimental study without a random selection of patients, the representativeness of the sample could be compromised, but the general data for our sample show results in concordance with other studies with frail patients [54]. As a future goal, it would be interesting to assess clinical outcomes after applying different proposals to individualise the therapeutic approach through a longitudinal follow-up study. Indeed, currently, there is a lack of evidence concerning this issue. A recent systematic review highlights the paucity of research into the impact of optimizing prescriptions on clinical outcomes in older people living with frailty [55]. However, there is some evidence that MR can reduce the number of hospital readmissions among older people [56] and can prevent the decline of mental health, with no significant effects on other outcome measurements, apart from a reduction in the number of prescribed medications [57,58]. Thus, overall it is accepted that several studies suggest that medication optimization could be safe, feasible well-tolerated, and lead to important benefits [55].

## 5. Conclusions

We can conclude that up to 90% of older people with multimorbidity presented at least one IP, regardless of their place of residence. However, after an individualized medication review, nursing-home patients presented a greater decrease in some pharmacological parameters related to adverse events, such as polypharmacy and therapeutic complexity, compared to those living at home.

We could consider nursing homes as a highly suitable scenario to carry out a periodic MR, due to its high prevalence of frail people and its feasibility to apply the recommendations of an MR.

Prospective studies with a robust design should be performed to demonstrate this quasi-experimental study along with a longitudinal follow-up on clinical outcomes.

## Figures and Tables

**Figure 1 ijerph-19-03423-f001:**
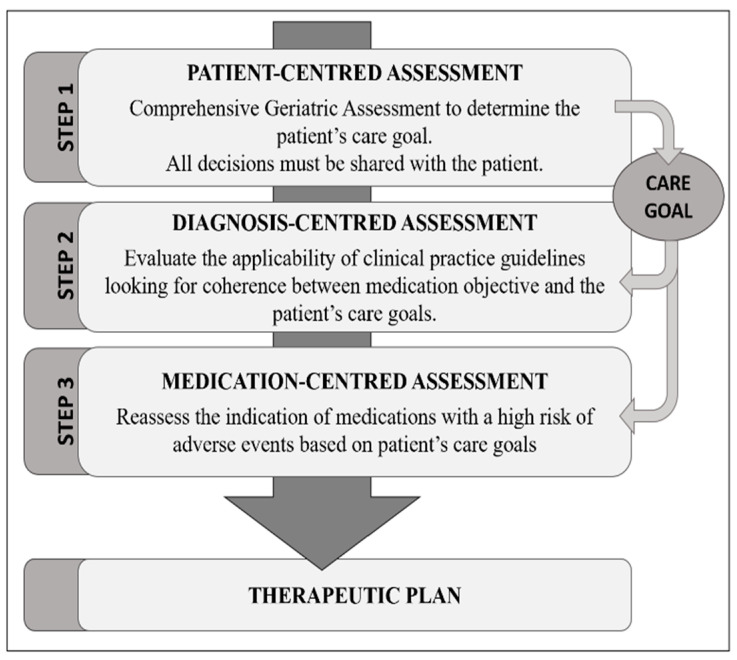
Patient-Centred Prescription (PCP) model.

**Figure 2 ijerph-19-03423-f002:**
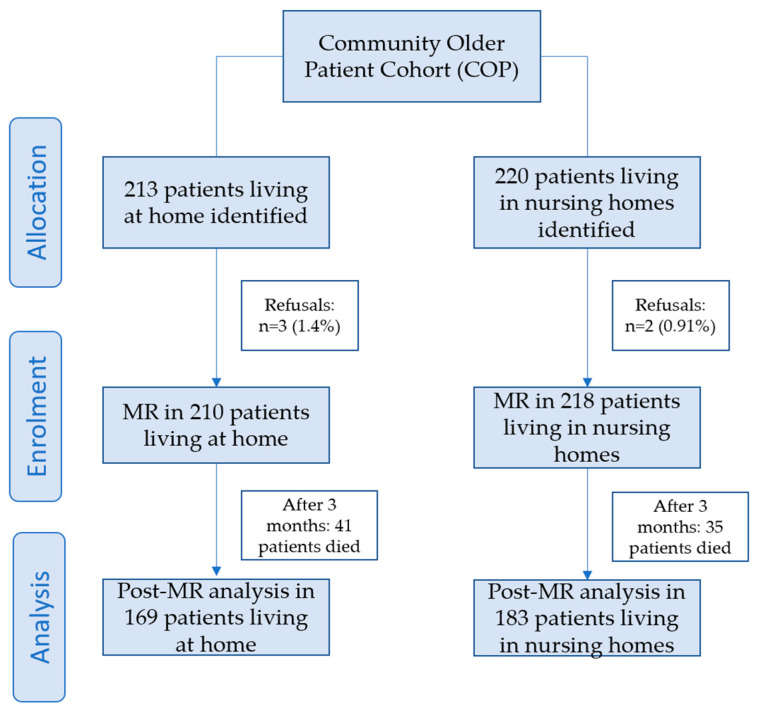
Number of patients pre- and post-medication review according to their place of residence.

**Table 1 ijerph-19-03423-t001:** HbA1c target according to each patient profile.

Target	Patients		
	Healthy Older Adults *	Frail Older Adults ^†^	Older Adults in a Probable EOL Situation ^‡^
Qualitative Glycaemic	Similar to those for diabetic young adults	Assess the decrease of therapeutic intensity	Quality of life preservation: avoid hypoglycaemic and symptomatic hyperglycaemic episodes
Quantitative HbA1c	≤7–7.5%	≤8.0%	Avoid reliance on A1C **
Therapeutic Goal ^††^	Prolong survival	Maintain functionality	Symptomatic treatment

* Good functional and cognitive status, and long-life expectancy. ^†^ With functional disability and dementia or moderately limited life expectancy. ^‡^ End-of-life (EOL) situation, understood as a period of 1–2 years. HbA1c, Glycosylated Haemoglobin. ** Glucose control decisions should be based on avoiding hypoglycaemia and symptomatic hyperglycaemia. ^††^ Based on the Patient Centred Prescription (PCP) Model.

**Table 2 ijerph-19-03423-t002:** COP-cohort’s baseline data according to their place of residence.

Baseline Data		Home *n* = 210 (49.1%)	Nursing Home *n* = 218 (50.9%)	*p*
Age (years), mean (SD *)		85.74 (6.74)	85.31 (8.48)	0.558
Gender	Men	83 (39.5%)	60 (27.5%)	0.09
	Women	127 (60.5%)	158 (72.5%)	
Medication management		57 (27.1%)	1 (0.5%)	<0.001
Barthel Index (BI), mean (SD)		65.21 (28.08)	35.21 (28.79)	<0.001
Functional status, BI ^†^ degrees	Independence: BI ≥ 95	46 (21.9%)	5 (2.3%)	<0.001
	Mild dependence: BI 90–65	75 (35.7%)	45 (20.6%)	
	Mod. dependence: BI 60–25	70 (33.3%)	59 (27.1%)	
	Severe dependence: BI ≤ 20	19 (9.0%)	109 (50.0%)	
Cognitive status	No dementia	75 (35.7%)	37 (17.0%)	<0.001
	Mild dementia (GDS 4)	29 (13.9%)	33 (15.1%)	
	Moderate dementia (from GDS 5 to GDS 6B)	66 (31.4%)	46 (21.1%)	
	Advanced dementia (from GDS 6C)	40 (19.0%)	102 (46.8%)	
Emotional status	Euthymic	102 (48.6%)	82 (37.6%)	0.02
	Depressive syndrome	93 (44.3%)	105 (48.2%)	0.421
	Anxiety syndrome	19 (9.0%)	16 (7.3%)	0.519
	Other psychiatric disorders	5 (2.4%)	29 (13.3%)	<0.001
Frailty Index (FI): VIG-Frail index, mean (SD)		0.34 (0.13)	0.43 (0.11)	<0.001
VIG-Frailty index degrees	No frailty (FI < 0.20)	30 (14.3%)	2 (0.9%)	<0.001
	Mild frailty (0.20–0.35)	69 (32.9%)	44 (20.2%)	
	Moderate frailty (0.36–0.50)	86 (41.1%)	115 (52.8%)	
	Severe frailty (FI > 0.50)	25 (11.9%)	115 (52.8%)	
End-of-life patients	No	168 (80.0%)	105 (48.2%)	<0.001
	Yes	42 (20.0%)	113 (51.8%)	
Number of geriatric syndromes, mean (SD)		2.78 (1.50)	3.07 (1.53)	0.047
Type of geriatric syndrome	Falls	76 (36.2%)	68 (31.2%)	0.274
	Dysphagia	36 (17.1%)	48 (22.0%)	0.204
	Pain	47 (22.4%)	52 (23.9%)	0.718
	Pressure ulcers	10 (4.8%)	10 (4.6%)	0.932
	Constipation	67 (31.9%)	68 (31.2%)	0.874
	Insomnia	106 (50.5%)	123 (56.4%)	0.218
	Malnutrition	16 (7.6%)	25 (11.5%)	0.176
	Incontinence	79 (37.6%)	153 (70.2%)	<0.001
	Previous delirium	32 (23.4%)	23 (44.2%)	0.007
Morbidities	Number of morbidities, mean (SD)	5.51 (2.21)	4.32 (1.94)	<0.001
	Age-adjusted Charlson Index, mean (SD)	3.37 (2.38)	3.15 (2.17)	0.33
Main therapeutic aim	Survival	31 (14.8%)	10 (4.6%)	<0.001
	Functional	128 (61.0%)	95 (43.6%)	
	Symptomatic	51 (24.3%)	113 (51.8%)	
Mortality		41 (19.5%)	35 (16.1%)	0.348

* SD: Standard Deviation. ^†^ BI: Barthel Index.

**Table 3 ijerph-19-03423-t003:** Baseline pharmacological data for the COP-cohort according to their place of residence.

Baseline Pharmacological Data		Home *n* = 210 (49.1%)	Nursing Home *n* = 218 (50.9%)	*p*
No. of med **	Mean (SD *)	8.84 (3.93)	7.45 (3.70)	<0.001
Polypharmacy	No polypharmacy: 0–4 med **	29 (13.8%)	51 (23.4%)	0.04
	5–9 med **	97 (46.2%)	108 (49.5%)	
	≥10 med **	84 (40.0%)	59 (27.1%)	
MRCI ^†^	Mean	33.12 (16.83)	28.44 (15.39)	0.03
DBI ^‖^	Mean (SD *)	1.08 (0.84)	1.26 (0.83)	0.031
IP ^‡^	Mean (SD *)	3.31 (2.42)	2.97 (2.10)	0.108
IP ^‡^	0 IP	25 (11.9%)	18 (8.3%)	0.209
	1 or more IP ^‡^	185 (88.1%)	200 (91.7%)	

* SD: Standard Deviation. ** med: medications. ^†^ MRCI: Medication Regimen Complexity Index. ^‖^ DBI: Drug Burden Index. ^‡^ IP: Inappropriate Prescription.

**Table 4 ijerph-19-03423-t004:** Pre-post medication review analysis according to their place of residence, of medication number, medication regimen complexity index (MRCI), anticholinergic and or sedative burden (DBI) and monthly drug expenditure (MDE).

			Home	Nursing Home	*p*
Medication number, mean (SD *)		Pre-MR	8.84 (3.94)	7.45 (3.70)	<0.001
		Post-MR	7.75 (3.58)	5.66 (3.58)	<0.001
		Difference	−1.20 (2.07)	−1.68 (1.84)	0.020
Polypharmacy N (%)	Pre-MR ^†^	No polypharmacy	29 (13.8%)	51 (23.4%)	0.004
		5–9 medications	97 (46.2%)	108 (49.5%)	
		≥10 medications	84 (40.0%)	59 (27.1%)	
	Post-MR	No polypharmacy	28 (16.6%)	77 (42.1%)	<0.001
		5–9 medications	91 (53.8%)	80 (43.7%)	
		≥10 medications	50 (29.6%)	26 (14.2%)	
MRCI ^‡^, mean (SD)		Pre-MR	33.1 (16.8)	28.4 (15.4)	0.003
		Post-MR	28.7 (14.9)	21.7 (14.4)	<0.001
		Difference	−4.8 (8.9)	−6.9 (7.4)	0.016
DBI ^‖^, mean (SD)		Pre-MR	1.08 (0.84)	1.26 (0.84)	0.031
		Post-MR	1.01 (0.78)	1.17 (0.86)	0.079
		Difference	−0.09 (0.35)	−0.13 (0.35)	0.359
MDE **, median (Q1; Q3)		Pre-MR	71.24 (29.6; 130.5)	51.81 (25.3; 99,1)	0.012
		Post-MR	60.59 (26.3; 122.2)	36.46 (17.7; 83,0)	<0.001
		Difference	−2.17 (−16.3; 2.0)	−6.41 (−16.9; −1.9)	<0.001

* SD: Standard Deviation. ^†^ MR: Medication Review. ^‡^ MRCI: Medication Regimen Complexity Index. ^‖^ DBI: Drug Burden Index. ** MDE: Monthly Drug Expenditure.

## Data Availability

The datasets generated during and/or analysed during the current study are available from the corresponding author on reasonable request.

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
