# Peer review of "Individualized Medication Review in Older People with Multimorbidity: A Comparative Analysis between Patients Living at Home and in a Nursing Home"

_ijerph, 2022, doi:10.3390/ijerph19063423_

Round 1

Reviewer 1 Report

The utilization of tailored mediation review for older individuals is an issue worth studying.  However, greater context clarification and data analysis are required.

Major concerns:

  1. In the introduction section, it stated: a) “Given the marked vulnerability of frail patients, there is concern and evidence they may not benefit from intensive management of chronic conditions in the same way as study populations do.”.  Please elaborate further on this statement by providing examples. b)“As a result, a medication review (MR) must be considered in frail older patients, which is a structured evaluation of patient’s medicines with the aim of optimizing medication use and improving health outcomes.” Please provide background information on current practice of “intensive management of chronic conditions about medication use”. Are there any formal evaluation?  If yes, please write the major differences between formal practice and the MR.  Please elaborate on the benefit of optimizing medication use on health outcomes.
  2. In the method section, please clearly stated the data collection method. Was all indices collected by reviewing the medical record only? Who performed the measurement?
  3. In the result section: a) In Table 2, it shows that only one elder had medication management at nursing home in the baseline, but how come the baseline pharmacological data was among 218 elders living at nursing home? How did you collect the baseline pharmacological data among elders living at nursing home? Pls. explain. b) No data on IP is shown on Table 4. c) For the pre-post medication review, this is better to have a table showing if the clinical outcomes improved according to their therapeutic aim type at nursing home. This may add to the evidence on the benefit of optimizing medication use on health outcomes.  Although you have mentioned this limitation at the discussion section, this is a very important analysis to show the goodness of PCP model that was suggested to apply in the nursing home.
  4. In the discussion section: as mentioned earlier, the longitudinal follow-up on clinical outcome is the future aim, however, the pre-post result on clinical outcomes should be disclosed even no significantly differences is found. No short-term benefit doesn’t imply no long-term benefit.

Author Response

Dra. Núria Molist-Brunet

Hospital Universitari de la Santa Creu de Vic

08500 Vic, Spain

nmolist@gmail.com

March 4th, 2022

Dear,

On behalf of all co-authors, I would like to thank you for allowing us to submit a revised version of our manuscript entitled Individualized Medication Review in Older People with Multimorbidity: A Comparative Analysis Between Patients Living at Home and in a Nursing Home (ref. ijerph-1604552) to be considered for publication in the International Journal of Environmental Research and Public Health.

The revision submitted includes detailed responses to the reviewers' comments and the manuscript has been edited accordingly, with track changes as you requested.

We hope that this revised version will be suitable for publication in the International Journal of Environmental Research and Public Health.

Thank you very much for this new opportunity.

Sincerely,

Núria Molist-Brunet

RESPONSE TO REVIEWER 1

The utilization of tailored mediation review for older individuals is an issue worth studying. However, greater context clarification and data analysis are required.

Major concerns:

  1. In the introduction section, it stated:

1. a) “Given the marked vulnerability of frail patients, there is concern and evidence they may not benefit from intensive management of chronic conditions in the same way as study populations do.”.  Please elaborate further on this statement by providing examples.

RESPONSE: In response to your suggestion, we have given an explanation along with some new references (highlighted in yellow) (see L056) (references 4-12 in the manuscript)

1. b) “As a result, a medication review (MR) must be considered in frail older patients, which is a structured evaluation of patient’s medicines with the aim of optimizing medication use and improving health outcomes.” Please provide background information on current practice of “intensive management of chronic conditions about medication use”. Are there any formal evaluation?  If yes, please write the major differences between formal practice and the MR. Please elaborate on the benefit of optimizing medication use on health outcomes.

RESPONSE: To clarify any possible confusion, we have written the following explanation:

Globally, a medication review (MR) is considered essential in frail patients, especially in those with polypharmacy [1,2]. The need to carry out a periodic MR in frail older adults is accepted, using a specific tool to consider parameters such as quality of life, functionality, main care goal, and life expectancy [1]. According to the Pharmaceutical Care Network Europe (PCNE), three types of MR, which is defined as a structural evaluation of the patient’s medicines to optimize their use and improve health outcomes might be distinguished: simple, intermediate, and advanced [3]. This fact entails detecting drug-related problems and recommending interventions to ensure individualized prescription. According to the PCNE’s definition,  the PCP model could be considered an advanced MR (based on medication history, patient information, and clinical information) [3,4]. This is the reason why we propose a study applying the PCP model as a type of MR. We have also added some references where the application of the PCP model shows the capacity to identify IP, reduce polypharmacy prevalence and improve  medication adherence in older patients [5–9].

In accordance with this explanation, we have made some amendments to this part of the Introduction section in order to make it more comprehensible (see L074, L083). We have added some new references, highlighted in yellow (references 13-18, 20 in the manuscript).

2. In the method section, please clearly stated the data collection method. Was all indices collected by reviewing the medical record only? Who performed the measurement?

RESPONSE: We realise that we need to clarify this process. To make it more understandable, below we have given a detailed explanation, and we have made some amendments to the manuscript (see L179):

Periodic meetings were scheduled between different primary care teams (General Practitioner and nurse) and a consultant team (a geriatrician and a clinical pharmacist). Before the meeting, the primary care team identified the older patients with multimorbidity who may be having difficulties with prescription management. Once the patient was identified, the primary care team informed the patient (or main caregiver in cases of incapacity) and requested their informed consent to carry out a medication review along with a consultant team.

If the patient consented, a global assessment of the patient and a medication review were carried out by the consultant team as a first step (before the interdisciplinary meeting). Then, during the meeting, the patient’s global baseline assessment and the medication review were made (the Barthel index evaluation was calculated and the Global Deterioration Scale and frailty index were evaluated). all prescribed medications were also recorded (and the related pharmacological indices were calculated: MRCI, DBI and monthly drug expenditure). Proposals were made for optimizing the pharmacological plan. Finally, both teams reached an agreement on the proposals for optimizing individual’s medication

All these data were transferred to a TeleForm document, which provides an automated optical reading. Patients’ data were incorporated successively during the different meetings.  Approximately 3-4 patients were incorporated in each meeting

After the meeting, the agreed medication proposals were shared between the primary care team and the patient during a conventional visit and, in accordance with the decisions made between them, the accepted proposals were recorded in the electronic prescription.

At three months of follow-up, the electronic prescription was reviewed by the consultant team, and the proposals recorded in the electronic prescription were considered to be accepted and were transferred to the TeleForm document.

3. In the result section:

a) In Table 2, it shows that only one elder had medication management at nursing home in the baseline, but how come the baseline pharmacological data was among 218 elders living at nursing home? How did you collect the baseline pharmacological data among elders living at nursing home? Pls. explain.

RESPONSE: We understand your query. As is usually the case, in the nursing homes in the study the residents did not manage their own medication, with the exception of one resident, who lives in a nursing home because he doesn’t want to live alone but all of whose cognitive and functional abilities remain intact. Thus, the baseline pharmacological data was collected during the interdisciplinary meeting using the electronic prescription and being supervised by the nursing home team (as we have explained in the answer to the previous query).

b) No data on IP is shown on Table 4.

RESPONSE: This is because we did not analyse the IP data post-MR. IP was only analysed during the meeting between the primary care and consultant teams. It might have been interesting to analyse it, but we did not carry out a second MR.

c) For the pre-post medication review, this is better to have a table showing if the clinical outcomes improved according to their therapeutic aim type at nursing home. This may add to the evidence on the benefit of optimizing medication use on health outcomes.  Although you have mentioned this limitation at the discussion section, this is a very important analysis to show the goodness of PCP model that was suggested to apply in the nursing home.

RESPONSE: We totally agree with your suggestion. Unfortunately, in our study a longitudinal follow-up on clinical outcomes was not carried out. It is one of the project’s weaknesses. Indeed, it is accepted in the literature that globally there is a lack of evidence regarding this issue. A recent systematic review highlights the paucity of research into the impact of optimizing prescription in older people living with frailty (cite reference number 58 in the manuscript). However, it accepts that several studies suggest that medication optimization could be safe, feasible, well-tolerated and lead to important benefits.

We firmly believe that it should be one of the next steps for us to take in the future.

4. In the discussion section: as mentioned earlier, the longitudinal follow-up on clinical outcome is the future aim, however, the pre-post result on clinical outcomes should be disclosed even no significantly differences is found. No short-term benefit doesn’t imply no long-term benefit.

RESPONSE: We regret to say that we did not in fact perform a longitudinal follow-up on any clinical outcome. Thus, in the “Discussion” section we can only include it as a limitation.

Furthermore, we propose expanding the information regarding this lack of studies in the “Discussion” section (see L366), with highlighted new references (references 55-58 in the manuscript)

REFERENCES

  1. Poudel A, Peel N, Mitchell C et al. A systematic review of prescribing criteria to evaluate appropriateness of medications in frail older people. Rev Clin Gerontol. 2014;24(4):304–18.
  2. Liau SJ, Lalic S, Sluggett JK, Cesari M, Onder G, Vetrano DL, et al. Medication Management in Frail Older People: Consensus Principles for Clinical Practice, Research, and Education. J Am Med Dir Assoc. 2021;22(1):43–9.
  3. Griese-Mammen N, Hersberger KE, Messerli M, Leikola S, Horvat N, van Mil JWF, et al. PCNE definition of medication review: reaching agreement. Int J Clin Pharm. 2018;40(5):1199–208.
  4. Thompson W, Lundby C, Graabæk T, Nielsen DS, Ryg J, Søndergaard J, et al. Tools for Deprescribing in Frail Older Persons and Those with Limited Life Expectancy: A Systematic Review. J Am Geriatr Soc. 2019;67(1):172–80.
  5. Molist Brunet N, Sevilla-Sánchez D, Amblàs Novellas J, Codina Jané C, Gómez-Batiste X, McIntosh J, et al. Optimizing drug therapy in patients with advanced dementia: A patient-centered approach. Eur Geriatr Med. 2014;5(1):66–71.
  6. Molist Brunet N, Espaulella Panicot J, Sevilla-Sánchez D, Amblàs Novellas J, Codina Jané C, Altimiras Roset J, et al. A patient-centered prescription model assessing the appropriateness of chronic drug therapy in older patients at the end of life. Eur Geriatr Med. 2015;6(6):565–9.
  7. Molist-Brunet N, Sevilla-Sánchez D, Puigoriol-Juvanteny E, González-Bueno J, Solà- Bonada N, Cruz-Grullón M, et al. Optimizing drug therapy in frail patients with type 2 diabetes mellitus. Aging Clin Exp Res. 2020;32(8):1551–9.
  8. Molist-Brunet N, Sevilla-Sánchez D, González-Bueno J, Garcia-Sánchez V, Segura-Martín LA, Codina-Jané C, et al. Therapeutic optimization through goal-oriented prescription in nursing homes. Int J Clin Pharm. 2021;
  9. González-Bueno J, Sevilla-Sánchez D, Puigoriol-Juvanteny E, Molist-Brunet N, Codina-Jané C, Espaulella-Panicot J. Improving medication adherence and effective prescribing through a patient-centered prescription model in patients with multimorbidity. Eur J Clin Pharmacol. 2021.

Reviewer 2 Report

To the Authors:

The manuscript is dedicated to issues of crucial significance in geriatric care: polypharmacy, inappropriate prescriptions and medication reviews. Results of studies in this area might be important for clinical practice and setting standards of care for older adults living in the community as well as institutionalized seniors. 

In the reviewer’s opinion, the manuscript requires some changes to make the presentation of the study and the results more comprehensible and therefore to increase the potential for applicability of the results in practice. Suggestions and questions are presented below:

TITLE 

When there is a comparison, conjunction “and” should be used. Therefore, a change in the manuscript title should be considered, i.e.: “Individualized Medication Review in Older People with Multimorbidity: A Comparative Analysis Between Patients Living at Home and in Nursing Homes.” A shorter title also might be considered.

ABSTRACT

Grammar check needed in lines 19 (“…leads to optimize…”) and 20 (“…on a cohort of older patients…”)

INTRODUCTION

Well written and concise. Grammar check needed in line 67 (“…on a cohort of older patients…”)

MATERIALS AND METHODS 

2.1. Design

According to the Authors, “…a cohort of patients who lived in the community…either in their own home or in a nursing home…” Please, justify why patients living in a nursing home (institutionalized) are viewed as people living in the community. It is usually accepted in the literature that the term “living in the community” or “community-dwelling” refers to people living in their homes.

line 81 – “to whom” is not grammatically correct

line 81-82 – the Authors explain that a General Practitioner identified difficulties with the prescription management and referred patients for a MR. There are however, several issues that need clarification:

  1. Patients living at home: How many GPs took part in the process of referral? How were they addressed by the research team? Were the GPs instructed about the criteria for identification of difficulties with the prescription management or was it GP’s subjective decision? How many patients were identified, how many refusals, exclusions to reach the final number of 210 patients? How many of them were included in the post-MR analysis?
  2. Patients living in nursing homes: Were the patients referred by GPs (not by the nursing home staff?)? How were nursing homes chosen for participation in the study? How many patients were identified, how many refusals, exclusions to reach the final number of 218 patients? How many of those were included in the post-MR analysis?

It would be very helpful to include a STUDY FLOW in a form of a graph, presenting the above data.

Moreover, the Authors should explain the difference between the exclusion criterion (line 84): “patients in their probable last hour or days of life” and the fact that “end-of-life patients” were included in the analysis (tab. 2). It is quite confusing without an explanation.

2.2. Data collection

There is a general rule, that the section ‘Methods’ should include a detailed description of data collected for the purpose of the study covered by the manuscript, but other data collected within the study but not presented in the section ‘Results’ should not be mentioned. Therefore, the Authors should explain the following issues:

  1. Geriatric syndromes (line 101) are not explained. In the section ‘Results’, table 2 (page 6) the following geriatric problems are included: falls, dysphagia, pain, pressure ulcers, constipation, insomnia, malnutrition, incontinence and previous delirium. These geriatric syndromes should be listed in the section ‘Methods’ with a detailed description on how these problems were diagnosed, e.g. falls – period in which falls occurred (6 months? 1 year?); pain – how pain was assessed? How pain was assessed in patients with dementia? Was a pain scale used?; constipation – definition or source of data; insomnia – definition or source of data; malnutrition – how was the diagnosis established (BMI?, nutritional status assessment scale?); previous delirium – time period? any episode suspected of delirium? Who provided diagnosis of delirium: GP?, family members? If there was not a unified definition for each of the above geriatric problems used for the whole study group, these data should be removed as they cannot be compared.
  2. Analytical data: several blood tests mentioned in lines 102-103: full blood count and electrolytes (other than calcium) were not included in the medication review process. The Authors should include these parameters in the ‘Results’ or skip this information in ‘Methods’.
  3. Diagnosis of depression is not mentioned in the section ‘Methods’, but included in the section ‘Results’. Was a clinical scale use to assess depressive symptoms?

Additionally, Medication Regimen Complexity Index and Drug Burden Index should be explained in more detail, because they are important outcomes of the study.

2.3. Medication Review

Were the GPs and medical staff of the nursing homes informed about the results of medication reviews and the therapeutic plan? In the description of MR only patients and caregivers are mentioned (line 140).

RESULTS

3.1. Subject Baseline Data

Line 205: the word “globally” is not the optimal choice as it is usually interpreted as “worldwide”. It would be more suitable to use another word e.g.: overall, in total.

Table2, page 5, last row: What does “Medication management” mean in relation to the patients and presented values(%)?

Table 2, page 6: emotional status and geriatric syndromes – Please, refer to the comments related to the section ‘Methods’ above.

3.2. Data of IPs

The title of this subsection seems not appropriate, because there is only one sentence related to IPs (line 222) and the section presents results of pre-post medication review.

Table 4. Why IP data were not included in the comparison of pre- and post medication review? In lines 221-222 the Authors state: “…there were no significant differences regarding IP according to the place of residence”. This finding should not prevent the Authors from presenting the change in the number of IPs within the ‘home’ and ‘nursing home’ groups pre- and post MR.

DISCUSSION

Discussion seems short for the amount of collected data. It would be valuable to address the following issues:

  1. Pre- and Post MR change of IPs within the groups, i.e. HOME pre and post, NURSING HOME pre and post (it will be possible after including the results concerning IPs in table 4).
  2. The authors should discuss the potential representativeness (or limitations of representativeness) of the study groups for the whole population of older adults living in the studied region as well as in Spain.
  3. The results of the study should be compared with results of other interventional studies conducted recently (rich body of evidence available).
  4. The role of communication between health care system representatives, e.g. GPs, nursing home staff, specialist care providers should be discussed in terms of the results of the study and literature search.
  5. The potential role of pharmacists in the process of collaborative medication review should be discussed.
  6. Feasibility and perspectives of introducing MR as a standard procedure in the health care system in Spain should be discussed. Examples of solutions existing in other countries should be considered.

CONCLUSIONS

The conclusions could be expanded with additional statements after expanding the section ‘Discussion’.

REFERENCES

Ref. No. 5, Authors: Molist-Brunet et al., 2021 seems to consider the same study group, study design and similar conclusions. Please, provide an explanation of the added value of the present manuscript. Additionally, it is not mentioned in the manuscript that the results of this study were 
already published elsewhere.

Ref. No. 30, Authors: Molist-Brunet et al., 2021 seems to consider partially the same study group, methodology and similar conclusions. Please, provide an explanation of the added value of the present manuscript. Additionally, it is not mentioned in the manuscript that the results of this study were already published elsewhere.

Seven out of 33 references were published more than 10 years ago. Please, consider providing more recent publications, if possible.

Reviewer 3 Report

This is a pre-post short study using a PCP model to identify potential inappropriate prescriptions and revise prescriptions accordingly.

This is an interesting study. The major flow is the organization of the text. The authors probably have some experience in filing relevant documents, but in this case there is a need of a flow (full elaborative text with minimal enumerations, and no bullet listing). There is also a mixed-up with methods and objectives at various levels, and a problem with the focus on the PCP or the actual results of the study. From 2.4, it may be concluded that the authors would like to test the need of PCP. Therefore, in my opinion there is a strong need of reorganization of the document.

Materials and methods should describe in order who/what, where, when, and how was assessed. A general description may be given in the beginning, but each type and group of data should be described in detail.

The authors should state the geographical area and number of cases that have been initially reviewed, hundreds, thousands? It is necessary to have an idea of the 428 cases in the general population or >65y. How many nursing homes were screened?

Then, separate paragraphs for each data set should be included, e.g. how the biochemical markers/material were collected. Were they retrieved from a database? Who performed the actual collection and evaluation? Then who, when, where, how evaluated Barthel index.

I suppose that once datasets were retrieved/collected, there was a team who performed the medical review. Is this the same team in all cases or were they separate teams in each facility?

In addition to the context, there is a need to rearrange tables and figures, use relevant and recent references, and provide a good discussion. To my understanding the major study outcomes include inappropriate prescriptions, number of drugs, Medication Regimen Complexity Index, Drug Burden Index, cost. The field is vast, so many comparisons to other studies may be given. Regarding the “intervention”, to which authors refer as proposal, it is now part of a standard medical protocol, and even pharmacists may change the regimens. The discussion needs to be focused around it. Comments on the population may be made, but better to be linked to the context, e.g. long-lived, thus greater need for regular drug evaluation. It is also not clear if this was a pilot study, if this is a part of a regular medical review protocol and if and how often it is suggested to run.

Overall, there have been many studies on polypharmacy and benefits of regimen change. Many of them include large number of participants, detailed lists of drugs and when longitudinal, they describe actual benefits on quality of life, morbidity and mortality. This is a very small study. The authors could make additions and improvements to the context. Considering the “intervention”, authors are also advised to comment on its value and applicability.

Please follow the suggested abstract format: structured, but without headings.

Explain IP early.

428 patients is not a result.

What “implemented optimization proposals” refers to?

The use of 1.1 is without meaning when there is no 1.2.

Please revise syntax and content of 1.1, as to make sense. A comparative analysis is a tool. What are the aims of the study? In addition, IJERPH scope includes public health. There is a need to extent somehow the introduction, and finally to set the objectives of the study under this scope. Please state the type of your study, and how it fits.

Then revise the beginning of the abstract and the manuscript accordingly.

Round 2

Reviewer 1 Report

The revised version has been greatly enhanced. It's unfortunate that data on clinical outcomes was not obtained at the post-MR period, further information on the positive influence of PCP-based MR on clinical outcomes will hopefully be available in the future. Just a remark that the explanations in the response letter were clearer than the highlighted sections in the amended version. Please double-check the revised version which should be aligned with the response letter.

Author Response

Dra. Núria Molist-Brunet

Hospital Universitari de la Santa Creu de Vic

08500 Vic, Spain

nmolist@gmail.com

March 9th, 2022

Dear,

On behalf of all co-authors, I would like to thank you for allowing us to submit a revised version of our manuscript entitled Individualized Medication Review in Older People with Multimorbidity: A Comparative Analysis Between Patients Living at Home and in a Nursing Home (ref. ijerph-1604552) to be considered for publication in the International Journal of Environmental Research and Public Health.

The revision submitted includes detailed responses to the reviewers' comments and the manuscript has been edited accordingly, with track changes as you requested.

We hope that this revised version will be suitable for publication in the International Journal of Environmental Research and Public Health.

Thank you very much for this new opportunity.

Sincerely,

Núria Molist-Brunet

RESPONSE TO REVIEWER 1

The revised version has been greatly enhanced. It's unfortunate that data on clinical outcomes was not obtained at the post-MR period, further information on the positive influence of PCP-based MR on clinical outcomes will hopefully be available in the future. Just a remark that the explanations in the response letter were clearer than the highlighted sections in the amended version. Please double-check the revised version which should be aligned with the response letter.

RESPONSE: We really appreciate your suggestions. We propose some amendments in the manuscript according to the response letter, highlighted in blue (see L074, L194, and L199)

Reviewer 2 Report

The Authors have properly addressed all issues raised in the first review. In the reviewer's opinion adding explanations in the manuscript made it more comprehensible to readers. There are, however, three issues to be considered before the decision to publish the manuscript:

  1. As stressed in the first review, the Authors should explicitly mention that the results of the same study and the same cohort (but different range of research data) were previously published. Such statement might fit in the section 1. Introduction or in section 2.1. Design.
  2. There is a mistake in Fig. 2 (line 247): the element on the bottom right refers to "183 patients living at home" while it should refer to: "183 patients living in nursing homes"
  3. Please, perform a grammar check of the newly added text, especially page 2, lines 53-65.

Author Response

Dra. Núria Molist-Brunet

Hospital Universitari de la Santa Creu de Vic

08500 Vic, Spain

nmolist@gmail.com

March 9th, 2022

Dear,

On behalf of all co-authors, I would like to thank you for allowing us to submit a revised version of our manuscript entitled Individualized Medication Review in Older People with Multimorbidity: A Comparative Analysis Between Patients Living at Home and in a Nursing Home (ref. ijerph-1604552) to be considered for publication in the International Journal of Environmental Research and Public Health.

The revision submitted includes detailed responses to the reviewers' comments and the manuscript has been edited accordingly, with track changes as you requested.

We hope that this revised version will be suitable for publication in the International Journal of Environmental Research and Public Health.

Thank you very much for this new opportunity.

Sincerely,

Núria Molist-Brunet

RESPONSE TO REVIEWER 2

The Authors have properly addressed all issues raised in the first review. In the reviewer's opinion adding explanations in the manuscript made it more comprehensible to readers. There are, however, three issues to be considered before the decision to publish the manuscript:

  1. As stressed in the first review, the Authors should explicitly mention that the results of the same study and the same cohort (but different range of research data) were previously published. Such statement might fit in the section 1. Introduction or in section 2.1. Design.

RESPONSE: In response to your suggestion, we amended it in the manuscript (see L117)

  1. There is a mistake in Fig. 2 (line 247): the element on the bottom right refers to "183 patients living at home" while it should refer to: "183 patients living in nursing homes"

RESPONSE: We apologize for it. It is corrected in the manuscript (see L263)

  1. Please, perform a grammar check of the newly added text, especially page 2, lines 53-65.

RESPONSE: It is corrected in the manuscript. See L026, L056-068, L088-089, L141-147, L198, L263, L284, L352, L359, and L369